# Reconstructing the Temporal Origin and the Transmission Dynamics of the HIV Subtype B Epidemic in St. Petersburg, Russia

**DOI:** 10.3390/v14122748

**Published:** 2022-12-09

**Authors:** Marina Siljic, Valentina Cirkovic, Luka Jovanovic, Anastasiia Antonova, Aleksey Lebedev, Ekaterina Ozhmegova, Anna Kuznetsova, Tatiyana Vinogradova, Aleksei Ermakov, Nikita Monakhov, Marina Bobkova, Maja Stanojevic

**Affiliations:** 1Institute of Microbiology and Immunology, Faculty of Medicine, University of Belgrade, 11000 Belgrade, Serbia; 2Institute for Oncology and Radiology of Serbia, 11000 Belgrade, Serbia; 3Laboratory of T-Lymphotropic Viruses, N.F. Gamaleya National Research Center of Epidemiology and Microbiology, 123098 Moscow, Russia; 4St. Petersburg City AIDS Center, 190103 St. Petersburg, Russia

**Keywords:** St. Petersburg, Russia, HIV subtype B, transmission clusters, phylodynamics, latent class analysis

## Abstract

The HIV/AIDS epidemic in Russia is among the fastest growing in the world. HIV epidemic burden is non-uniform in different Russian regions and diverse key populations. An explosive epidemic has been documented among people who inject drugs (PWID) starting from the mid-1990s, whereas presently, the majority of new infections are linked to sexual transmission. Nationwide, HIV sub-subtype A6 (previously called A_FSU_) predominates, with the increasing presence of other subtypes, namely subtype B and CRF063_02A. This study explores HIV subtype B sequences from St. Petersburg, collected from 2006 to 2020, in order to phylogenetically investigate and characterize transmission clusters, focusing on their evolutionary dynamics and potential for further growth, along with a socio-demographic analysis of the available metadata. In total, 54% (107/198) of analyzed subtype B sequences were found grouped in 17 clusters, with four transmission clusters with the number of sequences above 10. Using Bayesian MCMC inference, tMRCA of HIV-1 subtype B was estimated to be around 1986 (95% HPD 1984–1991), whereas the estimated temporal origin for the four large clusters was found to be more recent, between 2001 and 2005. The results of our study imply a complex pattern of the epidemic spread of HIV subtype B in St. Petersburg, Russia, still in the exponential growth phase, and in connection to the men who have sex with men (MSM) transmission, providing a useful insight needed for the design of public health priorities and interventions.

## 1. Introduction

Russia is home to over 140 million people, and the site of the fastest growing HIV/AIDS epidemic in Europe. Newly diagnosed infections in 2020 in the Russian Federation contributed 57% of all new cases in the World Health Organization (WHO) European Region, with the highest rate of 40.8 new HIV diagnoses per 100,000 population [1]. According to the latest official data, since 1987 until December 2020 nearly 1.5 million cases of HIV infection have been registered, whereas over 1.1 million people in Russia currently live with HIV [2]. In contrast to the global epidemic pattern, the HIV epidemic in Russia continues to expand significantly [3].

Nevertheless, HIV epidemic burden is non-uniform in different Russian regions, with the major metropolitan cities, Ural and Siberia, disproportionately bearing the burden: the most affected regions are Urals (Yekaterinburg-107398, Tyumen-62414), the Volga region (Samara-79442) and Siberia (Irkutsk-64554, Kemerovo-77971), as well as the largest cities, Moscow (67913) and St. Petersburg (69123) cases. An explosive epidemic has been documented among people who inject drugs (PWID), starting from the mid-1990s, whereas presently, the majority of new infections are linked to sexual transmission [1,2]. Nationwide, among the cases with known infection risk, about 65% reported heterosexual contact, about 2% are men who have sex with men (MSM), and about 32% are PWID, while male to female ratio among newly diagnosed cases is approximately 60:40, pointing to heterosexual risk, MSM, and PWID as the key populations in the context of the Russian HIV epidemic.

Regarding HIV subtype distribution, nationwide epidemiological surveys and a number of local studies indicated a large predominance of the A6 sub-subtype virus (previously called A_FSU_), with the significant presence of other subtypes, namely subtype B and CRF063_02A [4,5,6,7,8,9,10,11,12]. The abovementioned big HIV epidemic among PWID has been shown to be the result of two single introductions of sub-subtype A6 and subtype B (labelled also B_FSU_ or, IDU-B) [13,14].

The spread of subtype B seems to have gone through several independent viral lines and has circulated through sexual contacts, mainly among MSM at relatively low transmission rates, prior to its more significant presence in the overwhelming HIV epidemic in Russia [15,16,17].

The occurrence of HIV subtype B was considered rare in Russia; nevertheless, a recent analysis showed its significant presence in some regions, e.g., in the Russian Far East, probably linked to labor migrants [16]. Indeed, the population of HIV-1 subtype B strains in Russia and countries of the former Soviet Union has a complex structure. Besides numerous strains related to subtype B sequences circulating in Western Europe and seemingly originating from multiple independent introductions, most probably from Europe and the USA (“Western B”), there are at least two circulating subtype B viruses that are specific for this region. One of them is associated with the outbreak among PWID in Nikolaev, Ukraine, in 1996, namely IDU-B (= BFSU = FSU-B = B_FSU_) [14]. The initial origin of this strain has not been elucidated; it might be a recombinant of two or more subtype B variants. The circulation of subtype B variant related to the Brazilian subtype B has also been described, along with Thai subtype B’ virus [13].

In St. Petersburg, the first HIV-1 infections among PWIDs were diagnosed in the mid-1990s [18]. A number of studies so far have addressed subtype distribution and HIV molecular epidemiology in Russian’s second largest city, however, in depth phylogenetic studies are still scarce, especially regarding non-subtype A [19,20].

This study explores HIV subtype B sequences from St. Petersburg, collected from 2006 to 2020, in order to phylogenetically investigate and characterize transmission clusters, focusing on their evolutionary dynamics and potential for further growth, along with socio-demographic analysis of the available metadata.

## 2. Methods

### 2.1. Study Population

The study was approved by the “Health committee of the government of Saint Petersburg. Saint Petersburg state budget health institution Center for prevention and control of AIDS and infectious diseases”, Protocol No199, 2019. The study included HIV-1 subtype B infected, consenting patients, and referred to the St. Petersburg municipal AIDS center for genotypic drug resistance testing by DNA sequencing of partial *pol* gene. Blood samples were collected in the period 2006–2020, in the scope of routine laboratory monitoring. The eligibility criteria for participant selection included: HIV-1subtype B infected; seropositive adults (men and non-pregnant women) aged 18 years or above; with a place of residence in St. Petersburg; and viral load higher than 1000 copies per ml of blood.

Epidemiological, clinical, and behavioral data were collected using a standardized questionnaire. Transmission risk was categorized as heterosexual contact, MSM, PWID, transfusion, vertical transmission, or unknown.

### 2.2. RNA Isolation, RT PCR and DNA Sequencing

Plasma samples were genotyped using the ViroSeq HIV-1 Genotyping System (Abbott, Chicago, IL, USA) or AmpliSens^®^ HIV-Resist-Seq (InterLabService Ltd., Moscow, Russia) in accordance with the manufacturer’s instructions.

### 2.3. Phylogenetic Analysis

Before starting this phylogenetic investigation, dataset was carefully checked for the presence of multiple sequences from the same patent, longitudinally followed during the study period. In the final dataset only single sequence per patient was included. Sequence alignment was performed by the Clustal W algorithm implemented in the MEGA 10.2 software, followed by maximum likelihood (ML) phylogenetic reconstruction. (https://www.megasoftware.net/, accessed on 29 November 2022). Prior to further phylogenetic analysis, sequences were codon-stripped for major drug resistance-associated positions, to avoid differential selection pressure bias between treated and naïve patients. [21]. The best fitting nucleotide-substitution model for the representative dataset, as determined using model selection employed in the jModeltest software, was the general time reversible model of nucleotide-substitution with a proportion of invariant sites (i) and gamma distribution of rates (G) (GTR + I + G), selected according to the Akaike Information Criterion (AIC), by [22,23]. Bootstrap support for the tree nodes of the reconstructed phylogenetic trees were calculated with 1000 replicates. Genetic distance calculations and phylogenetic tree topologies was also confirmed under the ML algorithm in the Paup software. Phylogenetic trees were visualized using the FigTree program v 1.4.2 [24]. Bayesian phylogenetic analyses, estimating the posterior distributions of phylogenetic tree topologies, were also performed, using Markov chain Monte Carlo (MCMC) methods in the MrBayes software v 3.2.7 [25].

Transmission clusters investigation was performed through an analysis of a series of criteria sets as described previously, aimed to maintain specificity in identifying true transmission clusters, yet to avoid an underestimation of the incidence of transmission chains, in the view of an extended 15 years timeframe of sampling in the present study [26,27,28]. According to the first set of criteria phylogenetic clades consisting of three or more sequences, with the genetic distance of 4.5% or less, minimal bootstrap support of 90% and the statistical support of a posterior probability higher than 0.9 in the Bayesian maximum clade credibility tree (MCCT), were considered as transmission clusters. A second criteria set was based on bootstrap support over 75% and genetic distance of less than 4.5%. These phylogenetic clades were further confirmed by means of BLAST analysis (http://blast.ncbi.nlm.nih.gov accessed on 10 May 2022), with the purpose to identify highly similar sequences, based on score significance, not included in the initial dataset. In particular, for every sequence belonging to a cluster identified by the second criteria set, five most similar sequences identified using the BLAST tool were included in tree reconstruction as previously described [26,28]. In the posterior analyses, only those clades with no intermixing of foreign sequences downloaded using BLAST, were considered as transmission clusters. In total, 220 BLAST identified sequences were retrieved from the NCBI database and included in the tree reconstruction through phylogenetic analyses as described.

### 2.4. Estimation of the Evolutionary History and the Effective Number of Infections

Bayesian Markov Chain Monte Carlo (MCMC) inference was applied to estimate the effective population size and tMRCA of HIV subtypes B, using the BEAST v1.10.4 software [29,30]. Bayesian analysis was performed using the following parameters: uncorrelated lognormal relaxed molecular clock; GTR nucleotide substitution model; estimated base frequencies, and gamma distribution model + invariant sites for heterogeneity among nucleotide sites. The analysis was performed using Bayesian skyline plot as tree prior. The MCMC chain length was set at 2 × 10^8^, with a burn-in of 10%, which gave an effective sample size (ESS) of >200. The output was analyzed using TRACER v1.7.1, with 5% uncertainty in parameter estimates reflected as the 95% highest posterior density (HPD) [31]. The maximum clade credibility (MCC) tree was obtained with the TreeAnnotator software v1.8.3 and visualized in FigTree v1.4.2. [24].

### 2.5. Birth-Death Skyline (BDSKY] Analysis

For this investigation, aimed to explore effective reproductive number (Re) over time, the BEAST 2 software package v 2.6.5 was used [32,33,34]. The effective reproduction number (Re), denotes the average number of secondary infections caused by an infected person at a given time during the epidemic, where Re above 1 means that the number of cases are increasing and Re below 1 means that the epidemic will die out. As included sequences are sampled at different time points, Birth Death Skyline Serial model and lognormal prior for Re was chosen. Other priors were set based on prior knowledge of the trends of HIV epidemic and analyzed dataset, as follows: the rate of becoming uninfectious was set as a log-normal prior with M = 1.8 and S = 0.2 (95% CI 1.7–10.2 corresponding to the maximal natural infective period of 10 years in HIV patients), as reported previously [35], the sampling probability was estimated using a prior beta (1.0, 9999) corresponding to a minority of cases sampled, while the number of dimension for the reproductive number was set to 10 for the whole dataset and to five for the four analyzed transmission clusters. The number of dimensions was set according the space between the origin of the epidemic and the present time of the investigated clade, reflecting the number of changes of Re during the investigated period [32]. In order to plot the output of the BDSKY analysis, the R software with the bdskytools software package, available on GitHub, was used. (https://github.com/cran, accessed on 30 May 2022).

### 2.6. Latent Class Analysis

Latent class analysis (LCA) was performed in order to reveal hidden subgroups or ‘discrete classes’ of participants’ epidemiological, demographic, and clinical profiles, especially to identify how these characteristics of the patient’s affect transmission clustering patterns of the viral sequences. The R software with polytomous latent class analysis software package “poLCA” distributed through the Comprehensive R Archive Network, CRAN, was used [36]. Herein, we examined participants risk profiles regarding eight types of latent class indicators (categorical latent variable). Latent class indicators included: gender, age, transmission risk, date of first reactive HIV test (time-period of diagnosis), CDC stage at the time of diagnosis, CD4 count, city/country where HIV infection was most probably acquired, and the inclusion of sequence within clusters/network. The analysis was begun with a 1-class model and increased the number of classes in each subsequent model seeking to minimize both the BIC and the AIC value before these values increased with the addition of another class. With nreps = 10, the latent class with the lowest value of both BIC and AIC was chosen.

### 2.7. Statisticall Analysis

To illustrate the overall epidemiological context of HIV epidemics in St. Petersburg, surveillance data on HIV incidence for the period 1999–2020 for Russia and St. Petersburg were obtained from the publically available national data (Rospotrebnadzor Reports (HIV Infection in the Russian Federation). Available online: http://www.hivrussia.info/dannyepo-vich-infektsii-v-rossii/ accessed on 24 May 2022) and reports by St. Petersburg City AIDS Center, St. Petersburg, Russia (personal communication).

Linear regression and descriptive statistics were conducted using standard statistical methods.

## 3. Results

### 3.1. Study Population and HIV Incidence Trends

In total, 247 patients were included in the study. After exclusion of mulitple sequences per patient, the final sequence dataset contained partial *pol* gene sequences from 198 patients, single sequence per patient. GenBank accession numbers of the included sequences are given in the Appendix A. The majority of patients included in the study were on highly active antiretroviral therapy (HAART) 75.7% (150/198), at the time of study. The overall population was 93% (185/198) men, with a great majority, 69.7%, reporting sex with men as risk for infection. Regarding the overall study population, MSM was the most frequently reported risk for infection, in 65.1% of all study patients (129/198), followed by heterosexual contact in 25.7% (51/198). Intravenous drug use (IVDU) was reported by a very small proportion of the study population, 9% (18/198), and vertical transmission in one patient.

A linear regression analysis of HIV incidence data for Russia and St. Petersburg revealed differing epidemiological trends (Figure 1). The overall downward trend in HIV incidence was observed in St. Petersburg, whereas an upward trend in HIV incidence is seen on the national level. Furthermore of note, is the fact that up to 2012 the epidemiological situation in St. Petersburg was worse than in Russia, illustrated by higher levels of HIV incidence in the city.

### 3.2. Phylogenetic Investigation of Transmission Clusters

A phylogenetic analysis of the dataset encompassing 198 HIV subtype B sequences from Saint Petersburg, sampled from 2006 to 2020, revealed the presence of 17 transmission clusters and 16 transmission pairs (Figure 2).

In total, 54% (107/198) of analyzed subtype B sequences were found grouped in 17 clusters that accomplished predefined sets of criteria. Clusters ranged from three to 14 sequences, with collection dates within the span of 1–12 years, and median pairwise genetic distance of 2.3% (range 0.1–4.5%). All identified transmission clusters fulfilled a second, less stringent criteria set, whereas 10 clusters, containing 58.8% (63/107) of the total number of clustering sequences, were defined by the first, most stringent criteria set. Four transmission clusters were found with over 10 sequences. Furthermore, 15.6% (32/198) sequences were identified within transmission pairs. Importantly, confirmatory cluster analysis with sequences obtained through BLAST search showed very few sequences identified by BLAST analysis to intermix within the identified transmission clusters. Of note, except for a single sequence reported from Cyprus, virtually no foreign sequence was BLAST identified, all of them had been sampled in Russia.

### 3.3. Timing the Origin and the Effective Population Size

Using Bayesian MCMC inference, tMRCA of HIV-1 subtype B was estimated to be around 1986 (95% HPD 1984–1991), Furthermore, temporal origin was also estimated for four local transmission clusters with the number of sequences above 10. Considering estimated temporal origin for the clusters 1, 2, 3 and 4, it was found to be in the early two-thousands. The median tMRCA for clusters 1, 2, 3 and 4 were estimated at 2002 (95% HPD 1997–2006), 2003 (95% HPD 1997–2006), 2001 (95% HPD 1995–2007) and 2005 (95% HPD 2000–2007), respectively.

A BSP coalescent tree prior enables the estimation of the effective population growth from the analyzed data of infected subjects as it progresses from the origin of the epidemic through time. The BSP analysis of the whole subtype B dataset identified exponential growth in viral effective population size (correlating with effective number of infections and/or transmission opportunities) from the mid-nineties to the mid-noughties (2005–2006) followed by constant growth phase onwards (Figure 3).

Effective population growth in four analyzed transmission clusters showed similar patterns as for the whole dataset with exponential growth phase starting from the beginning of the 2000s followed by an asymptotic phase approaching the present time.

### 3.4. Estimation of the Effective Reproductive Number

An estimation of the Re in BEAST 2.4.6 software package was firstly performed on the whole dataset, including 198 subtype B sequences (Figure 3). Results revealed the increase in Re value above 1 in the late nineties with the highest Re reaching the value of 1.5 from the beginning of 2000 to 2005, followed by a sharp decrease, however still reaming the value of around 1 till the end of the analyzed period. Secondly, Re was estimated for four largest transmission clusters revealing the value above 1 in all investigated clusters since their origination, followed gradually by decline. Of note, all four transmission clusters had a value above 1 approaching the present time (Figure 4).

### 3.5. Latent Class Analysis

LCA, performed aimed to correlate results of transmission clustering patterns obtained by phylogenetic analysis with available public health and clinical determinants, revealed the presence of four latent classes, based on AIC and BIC values (Figure 5).

Large transmission clusters, containing > 10 sequences, were dominantly present in the first latent class, which encompassed male subjects with sexually acquired HIV infection, predominantly MSM, late presenters aged between 30–50, with low CD4 cell count (below 100), and HIV diagnosed in the last 10 years of the study period. Notably, around one third of patients in this latent class reported HIV infection most probably acquired outside Saint Petersburg. The second latent class encompassed predominantly younger (under 30) MSM subjects, with HIV diagnosis in the preceding five years and presenting in the early stage of infection. The second latent class was the one encompassing smaller transmission clusters with the number of sequences up to 10. The third latent class encompassed an equal proportion of men and women, PWIDs and reporting heterosexual transmission risk for HIV infection, with the time of diagnosis dispersed across the whole study period (2006 to 2020) and mostly presenting at a late stage of HIV disease. The fourth latent class encompassed male subjects aged over five, reporting MSM risk and presenting in the late stage of HIV infection, with diagnosis of HIV infection established during the first 10 years of the study period.

## 4. Discussion

Here, we present the study that integrates an analysis of viral phylogenetics and analysis of socio-demographic metadata regarding HIV subtype B spread in St. Petersburg, Russia. Phylogenetic methods can provide unique insights in the transmission networks and the spread of the virus and have been used worldwide to map local HIV epidemics in correlation with transmission pathway, drug resistance, risk behavior, and cluster size [26,28,37]. Many studies were based on partial *pol* gene phylogeny, which has shown to be adequate to infer transmission events and to characterize epidemiological patterns of public health relevance [38]. When coupled to metadata analysis, they may give a quantitative description of transmission networks, by identifying socio-demographic correlates of clustering [28,35,39]. Furthermore, detailed analysis of HIV burden distribution may prompt targeted epidemiological analyses and preventive measures to reduce epidemic spread [40,41,42]. Insight into overall HIV incidence trend in St. Petersburg, and at the national level, reveals an opposing trend—a downward trend in HIV incidence in St. Petersburg vs. growing HIV epidemic in Russia. Historically, the first cases of HIV infection in Russia were registered in St. Petersburg long before the start of a large-scale epidemic. This early epidemic phase was linked to African students enrolled at the universities of the city at the time, that were since mostly lost to follow-up (either left the country or died), and the underlying HIV subtypes were not determined. Over the next years, certain cases were identified related to the introduction from Europe and the United States, all associated to MSM transmission and caused by HIV-1 subtype B [20,43]. Later, a large-scale outbreak of sub-subtype A6 occurred, mainly among PWIDs, overshadowing the continuing “hidden”, predominantly subtype B spread among MSM [43]. In the last decade, a changing pattern of HIV subtype distribution has been observed, involving all key populations [9,17]. This phenomenon required a deeper study, since the knowledge of the molecular structure of the epidemic as a whole and in separate groups gives grounds for the development of aimed preventive measures.

Previous studies have depicted transmission clusters in HIV spread in different regions of Russia [9,10]. To the best of our knowledge, the present study is the first one to address transmission clusters of HIV subtype B in St. Petersburg. In addition to 15.6% of analyzed subtype B sequences forming transmission pairs, our results imply that 54% of analyzed subtype B sequences, collected within the time span of 15 years, contributed to 17 clusters (of three sequences at minimum), similar as reported in comparable studies elsewhere [26,28,44,45,46,47]. Notably, confirmatory cluster analysis implies a completely autonomous Russian epidemic, without any significant intermixing of foreign sequences obtained through a BLAST search into the designated transmission clusters. The estimation of the effective reproductive number (Re) for the whole subtype B dataset, as well as for the four largest transmission clusters (containing 10 or more sequences), revealed a steady value above 1 in all investigated clades, implying a permanent increase in population size. In spite of the more or less pronounced gradual decline after the mid-noughties (2005–2006) all the four transmission clusters retained Re value above 1 until the end of the study period.

An estimation of the time of origin of HIV-1 subtype B in St. Petersburg, within the presented study, was to be around 1986 (95% HPD 1984–1991), in line with historical data and similar to the estimate regarding subtype B epidemic in Moscow [9]. This estimate coincides with major events of sociopolitical changes and turbulence in the former Soviet Union and Russia, when travel to Europe and USA, as well as contacts with the population of these countries, increased significantly. At the same time, other social events took place during “perestroika”, in particular an increase in illicit drug use, which, apparently, contributed to the spread of subtype B virus from Ukraine to Russia, and also to the subsequent formation of the CRF03_AB recombinant. Regarding the four large (over 10 sequences) subtype B St. Petersburg clusters, their origin has been estimated to be more recent, ranging from 2001 to 2005. This can be attributed to the changing pattern of HIV epidemic—notably, the explosive spread of HIV in Russia has been linked to illicit drug use, an epidemic predominantly caused by HIV sub-subtype A6. However, recent data imply an increasing dominance of sexual transmission, especially among newly diagnosed patients [1,2]. Notably, St. Petersburg is the first urban metropolis in the Russian Federation where a steady decline in new HIV infections has been recorded [48]. Nationwide, a concentrated HIV epidemic among people who inject drugs tends to dissolve to the generalized one, with prevalence in some regions reaching alarming heights [2]. However, through the City AIDS Center, St. Petersburg is among the few cities in the Russian Federation that has provided harm reduction programs, albeit on a limited scale [48]. With over 900 people living with HIV (PLWH) per 100,000 population, St. Petersburg is still among the Russian regions with the highest HIV prevalence, nevertheless, the incidence rate in the city is decreasing. Changing practices have been described among PWIDs in St. Petersburg, diminishing the risk of ongoing HIV transmission in this key population [49]. A recent study has shown the attribution of the transmission pathway with different HIV subtypes in St. Petersburg, with subtype B epidemic being linked to MSM transmission [43]. In the presented study, the overall population was 93% men, with over two thirds reporting sex with men as the risk for infection, while intravenous drug use was reported by a very small proportion of the population, 9%. At present, in Russian society, the disclosure of MSM infection risk is linked to substantial social stigma, even personal risk, making the studied dataset rather unique. Hence, complex metadata analyses may provide valuable insight into social networks implicated in HIV transmission and give grounds to planed preventive interventions. We performed latent class analysis, as an approach to reveal non-evident subgroups or ‘discrete classes’ of participants’ epidemiological, demographic, and clinical profiles within the study population, and correlate those with clustering pattern. Through this approach, four latent classes have been identified, with clearly distinguished social and epidemiological features. Two of the four identified latent classes encompassed mostly men with MSM transmission risk and were linked to transmission clusters. Larger clusters were associated to middle aged subjects with epidemiological links outside the city and more advanced HIV disease, whereas smaller clusters were associated to younger men with early stage HIV disease. In conjunction with the obtained results of phylodynamic and temporal analysis of transmission clusters, this points to the rising local spread of HIV subtype B among younger MSMs, who tend to present to care at earlier disease stage, along with maintaining continual spread within an older age group and more advanced HIV disease.

Of note, in view of the registered number of HIV cases in St. Petersburg of over 60,000, the sample size of the studied dataset may seem limited, and thus present as possible study limitation. However, in this regard, the documented low prevalence of subtype B in Russian epidemic should be considered, notably about 4% in the northwestern federal distric, where St. Petesburg is the main urban center [50]. Further, in Russia, the standard of care for HIV patients does not include genotyping prior to therapy initiation. Even in the case of therapy failure, access to genotyping is available for around 10% of patients in need of this analysis. Thus, the present study encompasses the total of all HIV subtype B sequence data, which have been obtained since genotypic resistance testing has been implemented in clinical care in St. Petersburg.

In conclusion, the results of our study imply a complex pattern of the epidemic spread of HIV subtype B in St. Petersburg, Russia, still in the exponential growth phase and in connection to the MSM transmission, highlighting the key population of young MSM as drivers of HIV subtype B epidemic, and providing a useful insight needed for the design of public health priorities and interventions. 

## Figures and Tables

**Figure 1 viruses-14-02748-f001:**
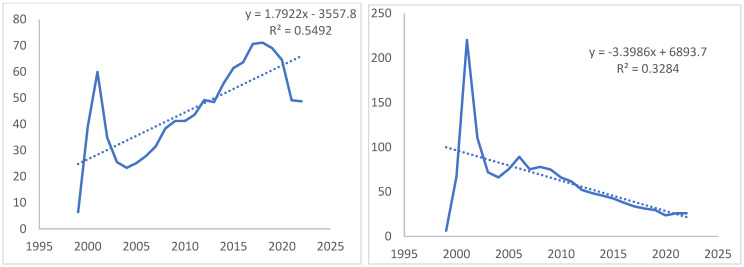
Linear regression trend of HIV incidence on 100,000 inhabitans in Russia (**left**) and St. Petesburg (**right**).

**Figure 2 viruses-14-02748-f002:**
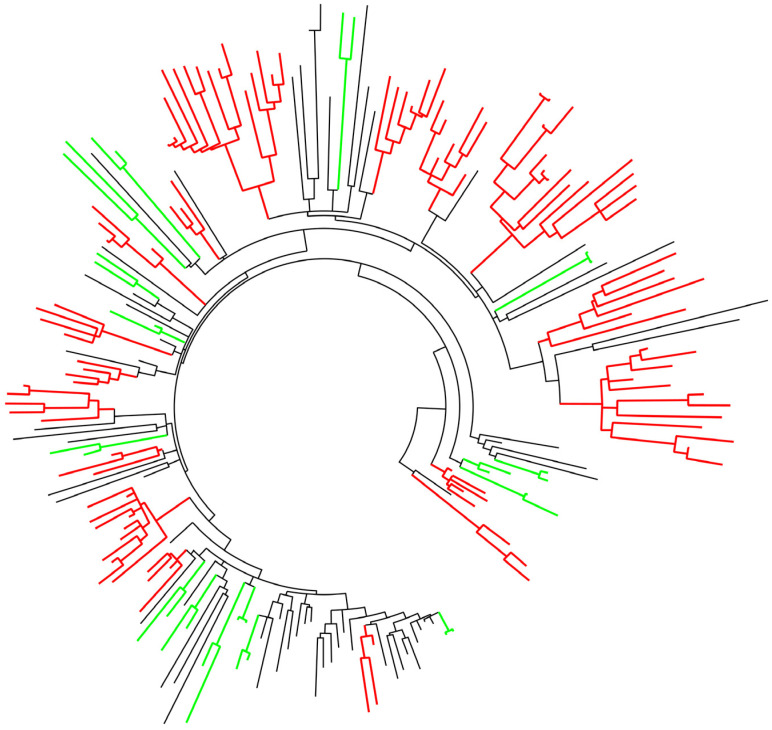
ML phylogenetic tree based on 198 HIV subtype B pol sequences sampled from 2006 to 2020 in St. Petersburg; tree was constructed in MEGA software under the GTR + G + I nucleotide substitution model. Transmission clusters are labeled in red. Transmission pairs are mark in green.

**Figure 3 viruses-14-02748-f003:**
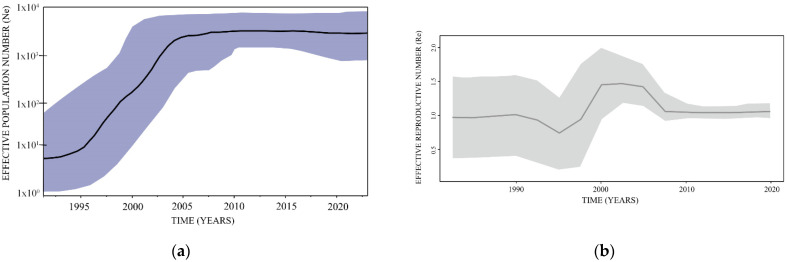
Bayesian skyline plot, representing estimation of the effective population growth of HIV subtype B in St. Petersburg (based on the ‘relaxed clock’ coalescent framework analysis on 198 sequences included in the study, in BEAST 1.10. software) (**left**); Effective reproductive number of the studied dataset of HIV subtype B in St. Petersburg, based on birth date skyline serial prior and 10 dimensions of Re in BEAST 2 software. Re plot was obtained in R software under the bdskytool software package (**right**).

**Figure 4 viruses-14-02748-f004:**
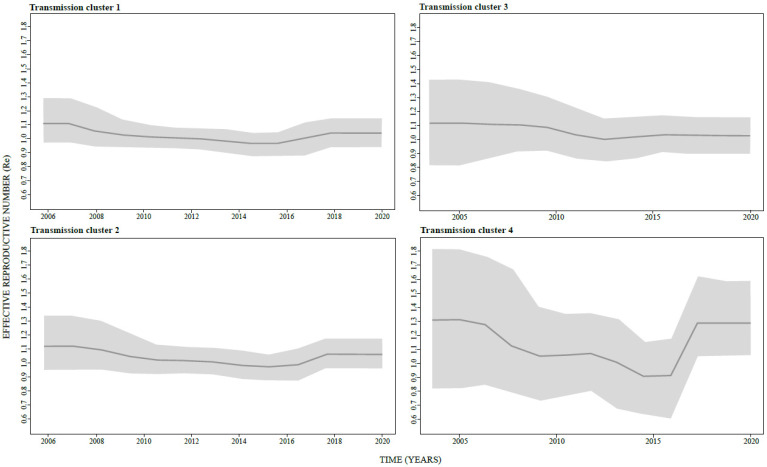
Effective reproductive number of four large transmission clusters (>10 sequences) obtained by birth death skyline analysis performed in BEAST 2 software under the following parameters; relaxed lognormal clock, birth death skyline serial prior, reproductive number with five dimensions. Re plot was obtained in R software under the bdskytool software package.

**Figure 5 viruses-14-02748-f005:**
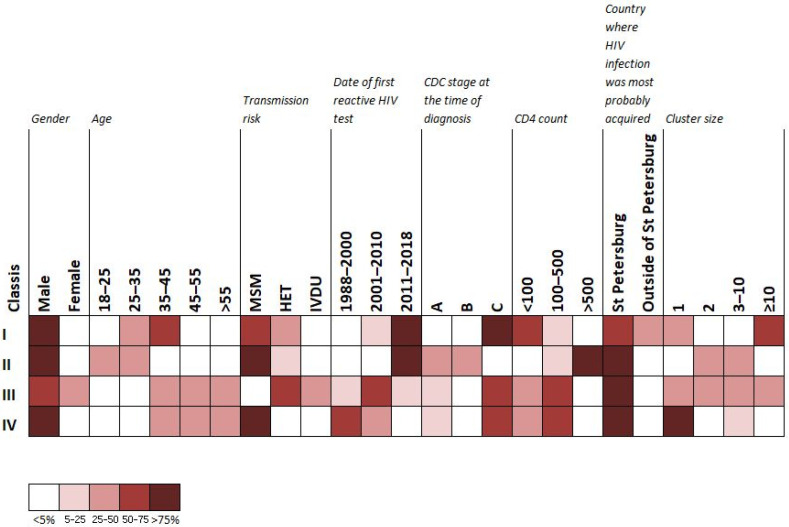
LCA result highlighted existence of four latent classes (indicated as I, II, III, IV), LCA was performed with poLCA software package in R, with study metadata and results of transmission clusters analysis as input. The distribution of participant characteristics within the latent class and predicted proportions in each cluster size by latent class are shown. Darker colors represent higher proportion as shown in the legend.

## Data Availability

NCBI accession numbers of the analyzed sequences are available as Appendix A.

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
