# Peer review of "Reconstructing the Temporal Origin and the Transmission Dynamics of the HIV Subtype B Epidemic in St. Petersburg, Russia"

_viruses, 2022, doi:10.3390/v14122748_

Round 1

Reviewer 1 Report

The work is devoted to the study of the introduction and further spread of HIV-1 subtype B in St. Petersburg. A number of studies have shown that subtype B is the second most common viral variant in the European part of the Russian Federation. Also, subtype B in the country is often associated with homosexual transmission. Men who have sex with men are a vulnerable group subject to severe discrimination in Russia. Some authors point out that the spread of HIV among homosexual men in Russia is a hidden and greatly underestimated epidemic. In this regard, this work is relevant and has scientific and practical significance.

In general, the work is assessed positively by the reviewer, however, there are a number of remarks and comments that can improve the quality of the work.

1. In section 3.2, the authors indicate that they search for closely related sequences using BLAST to confirm the results of cluster analysis. However, neither in the Materials and Methods section nor in Section 3.2 they describe the methodology and results of the search. This is an important aspect influencing the conclusions of the study, so it is necessary to describe the results of using BLAST in more detail.

2. Names and coordinates of the axes of Figure 2 in low resolution. It is desirable to improve the quality of the Figure.

3. How representative is the study sample? More than 50 thousand HIV-infected citizens live in St. Petersburg. If subtype B is present in 5-10% of the population, this means that 2.5-5 thousand people living with HIV live in the city. During the entire period of the epidemic, there could have been twice as many. In the opinion of the reviewer, the discussion section should indicate the limitations of the study related to the sample size.

Reviewer 2 Report

Manuscript ID           viruses-1866446

Title:   Reconstructing the temporal origin and the transmission         dynamics of     the HIV subtype B epidemic in St Petersburg,       Russia

Authors:     Siljic et al.

In this manuscript, Siljic et al. report on the phylogenetic characterization of 198 HIV subtype B sequences from St. Petersburg, Russia, collected from 2006 to 2020. The focus is on evolutionary dynamics and integrates viral phylogenetics and the analysis of socio-demographic data using latent classes. The Authors identified 17 clusters and 16 transmission pairs. They conclude the HIV epidemic in St. Petersburg is still in exponential growth and is mainly connected to MSM transmission.

In general, this manuscript is an interesting subject. The research method is well thought, and the interpretation is adequate and in accord with their results.

However, the Reviewer found several issues in the manuscript and would like to see them addressed by the Authors. Additionally found a major issue regarding the absence of ethics declarations. These issues must be addressed before this manuscript is ready to be considered for publication.

 Specific comments and suggestions:

 A.    INTRODUCTION

In general, the Introduction could use additional content. For readers not familiar with Russia and the situation of its HIV epidemic, it would be useful to elaborate on some points.

1. In paragraph 2: Authors mention “Nevertheless HIV epidemic burden is nonuniform in different Russian regions (with the major metropolitan cities, Ural and Siberia disproportionately bearing the burden) and diverse key populations.”. It would provide context if the authors could provide some details on this burden. Examples of which metro cities, what are the number of cases in different regions, etc. Could the Authors please specify what do they mean by “key populations” in their context.  

2. The Authors indicate the “majority of new infections are linked to sexual transmission”. Do they mean heterosexual transmission, or MSM or both? This statement can be made clearer.

3.    In paragraph 3, last 2 lines: Authors mention “introductions of sub-subtype A and subtype B”. Do the Authors mean “subtype A” or “sub-subtype A6”. Please clarify.

4.     In Paragraph 5: The Authors indicate the circulating strain of HIV-1 subtype B is not related to other strains circulating in Western Europe. Then, to which circulating strains is it related or more closely related? From which countries or regions? If it is not from Western Europe, then, from where?

B.    METHODS

Section 2.1

1.The study analyzed 198 HIV-1 B-subtype sequences. Could you elaborate: from a total of how many sequences that were available at your site. How were they selected?

2.     Please indicate which were the inclusion/exclusion criteria.

3.   Please elaborate, how the Authors made certain there were no repeated patient samples taken at different times (months or years). This would be important since the virus from a single patient tested with 15 years of separation may appear to be part of a cluster given the long interval of time being studied.

4.  Since partial pol gene sequences were used, please indicate the nucleotide positions relative to reference HXB2, which genes or regions included (protease, RT, integrase?) and which amino acid numbers were covered.

5.  Are these sequences submitted to GenBank? Which are the accession numbers?

6.  The Authors indicate that there was informed consent from patients however there is no ethics statement or indication whether the study was reviewed and approved by an ethics committee or internal review board.

Section 2.1

1.  Please elaborate on the methods. Which instrument(s) was (were) used to produce and analyze the sequences.

Section 2.5

1. The Authors indicate that “the number of dimension for the reproductive number was set to10 for the whole dataset and to 5 for the four analyzed transmission clusters.” Please clarify why the different numbers were used in the analysis.

Section 5.6

1.  Please review first sentence for clarity.

 C.    RESULTS

Section 3.2

1.  At the end of the section, the Authors indicate that, “virtually no foreign sequence was BLAST identified”. Please indicate then, which ones were identified? from where? Wouldn’t this give some indications to argument from where the HIV-1 subtype B was introduced into Russia?

Section 3.3

1.  Please correct at the end of paragraph (penultimate line), should read “beginning” and not “begging”.

Section 3.4

1.    Figure 3: Would it be possible to make the scale on the y-axis the same for the four clusters to allow for direct visual comparisons among them? Also make sure all lines are black and not dark gray.

D.    DISCUSSION

1.     According to the results, the epidemic still shows exponential growth. Can the Authors predict whether the number of infections with HIV-1 subtype B will grow to a point where it becomes the majority of the HIV cases or even surpass the current sub-subtype A6?

2.    The Authors indicate the estimated origin of HIV-1 subtype B in St. Petersburg to be around 1986, an estimate similar to one for Moscow. This timepoint (circa 1986) coincides with major events with the former Soviet Union and Russia. Could the Authors comment whether these might be somehow related or have influenced the introduction of HIV-1 subtype B into Russia?

 E.    REFERENCES

1.   The Authors should maintain consistency in the format for their references. Since the complete list of authors is included for most of the references, consequently the complete list of authors must be completed for references #15, #16 and #33, instead of writing just the first three authors and then et al. Please review.

 Major issue

Authors mention patients giving informed consent however there is no statement that the research was reviewed and approved by an Ethics Committee or equivalent. Also, there is no statement of funding source, no statement on Data Availability or Conflicts of Interest for the authors. Additionally, the Reviewer did not find any statements regarding the level of contribution of all the Authors to conducting this research and to the preparation of this manuscript.
